# Effect of Nebulized BromAc on Rheology of Artificial Sputum: Relevance to Muco-Obstructive Respiratory Diseases

Krishna Pillai [1,2], Ahmed H. Mekkawy [1,2], Javed Akhter [1,2] and David L. Morris [1,2,3,*]

1   Mucpharm Pty Ltd., Sydney, NSW 2217, Australia
2   Department of Surgery, St George Hospital, Sydney, NSW 2217, Australia
3   St George & Sutherland Clinical School, University of New South Wales, Sydney, NSW 2217, Australia
*   Correspondence: david.morris@unsw.edu.au; Tel.: +61-(02)-91132590

**Highlights:**

**What are the main findings?**

- BromAc is a better mucolytic compared to bromelain or N-acetylcysteine alone.
- BromAc reduces the viscosity and increases the flow rate of mucin.

**What is the implication of the main finding?**

- This ex vivo study suggest that testing BromAc in pre-clinical and clinical studies is warranted.
- BromAc is a potent mucolytic for respiratory disease patients.

**Abstract:** Respiratory diseases such as cystic fibrosis, COPD, and COVID-19 are difficult to treat owing to viscous secretions in the airways that evade mucocilliary clearance. Earlier studies have shown success with BromAc as a mucolytic agent. Hence, we tested the formulation on two gelatinous airway representative sputa models, to determine whether similar efficacy exist. Sputum lodged in an endotracheal tube was treated to aerosol N-acetylcysteine, bromelain, or their combination (BromAc). After measuring the particle size of aerosolized BromAc, the apparent viscosity was measured using a capillary tube method, whilst the sputum flow was assessed using a 0.5 mL pipette. Further, the concentration of the agents in the sputa after treatment were quantified using chromogenic assays. The interaction index of the different formulations was also determined. Results indicated that the mean particle size of BromAc was suitable for aerosol delivery. Bromelain and N-acetylcysteine affected both the viscosities and pipette flow in the two sputa models. BromAc showed a greater rheological effect on both the sputa models compared to individual agents. Further, a correlation was found between the rheological effects and the concentration of agents in the sputa. The combination index using viscosity measurements showed synergy only with 250 µg/mL bromelain + 20 mg/mL NAC whilst flow speed showed synergy for both combinations of bromelain (125 and 250 µg/mL) with 20 mg/mL NAC. Hence, this study indicates that BromAc may be used as a successful mucolytic for clearing airway congestion caused by thick mucinous immobile secretions.

**Keywords:** bromelain; acetylcysteine; BromAc; respiratory diseases; cystic fibrosis; COPD; COVID-19; drug repurposing

## 1. Introduction

In a healthy state, secretions in the respiratory tract are continuously cleared by the cilia lining the air passages and expectorated as sputum. However, the performance of this secretion is both dependent on the constituency and rheology of the secretion which affects ciliary clearance [1,2]. Viscosity in the range of 12–15 Pa·s and an elastic modulus of 1 Pa are necessary for optimal mucociliary clearance [3,4] Since the airways are continuously exposed to dust particles, pathogens and other exogenous molecules, the secretion propelled by ciliary motion serves to expel these harmful molecules continuously [5], and any

disruption may result in accumulation leading to respiratory infection, pneumonia, etc. [6]. In cystic fibrosis, a genetic disorder of the CFTR (cystic fibrosis transmembrane conductance receptor) gene, thick mucinous secretion accumulates in the respiratory tract, resulting in mucus plugging, bacterial infection and progressive deterioration in lung function [7]. A similar accumulation of mucinous secretion with bacterial infection may result in chronic obstructive pulmonary disease (COPD) and other respiratory infectious diseases [8]. Hence, stasis of airway mucus secretion may lead to a variety of respiratory disorders.

Normal airway secretion is composed of mucins (MUC5B and MUC5AC), water, sodium chloride, bicarbonate, and cellular materials with a pH of 7.0 [9–11]. The mucins serve as a protective barrier for the epithelial cells lining the respiratory tract as well as tissues beneath. In a diseased state, such as bacterial infection, owing to inflammation, excess mucin is secreted by the goblet cells as a protective measure with infiltrating white blood cells [12], accumulation of DNA fragments from dead cells and other solids along with sodium, chloride and bicarbonate imbalance [13] that results in loss of water with the secretion becoming thick and purulent (bacterial infection), which eventuates in the reduction or absence of ciliary clearance [14]. Acidic pH in cystic fibrosis sputum, as reported in some studies [15], may encourage the cross linking of DNA, mucins and cellular fragments with additional milieu for bacterial growth and sputum stasis [16].

Recent studies have shown a great similarity between the respiratory secretions of COVID-19 patients and cystic fibrosis (CF), indicating that successful treatment to mobilize the sputum in CF may be used for treating COVID-19 [17]. Further elevated levels of solids including proteins, hyaluronic acid, double stranded DNA (dsDNA) and infiltrating cells along with bacterial colonization were observed in both the diseases as compared to normal sputum, without significant changes in alkalinity (pH 7–8), although some studies on CF have shown a depressed pH [15]. Elevated levels of solids have also been shown to have a considerable effect on the rheology of the secretion and hence its clearance. Noticeably, the concentration of dsDNA is much higher (about 600 µg/mL) as compared to hyaluronic acid (about 7.0 µg/mL) in COVID-19 lung secretion, suggesting that dsDNA may have a higher impact on the rheology of the secretion.

Treatment options to enhance the clearance of stagnating sputa in CF include rehydration using saline [18], agents such as acetylcysteine (NAC), L-cystine, cysteamine and other pharmaceutical agents that are mucolytics [19–21], together with antimicrobial agents when there is infection [22]. In diseases such as cystic fibrosis, if stasis of the secretion in the respiratory tract can be halted or modulated, then the invasion of microbes may be reduced or avoided [23]. Similarly, in diseases such as pneumonia, COPD and CF, the prevention of mucociliary stasis may enable the prevention of microbial invasion, since stasis leads to accumulation of foreign agents, including microbes. Prevention of respiratory secretion stasis in COVID-19 may prevent deterioration in lung function, transplantation and death. Stasis of thick pulmonary secretions prevents oxygenation, with rapid respiratory failure in COVID-19 patients [24,25]. Further, with current development of various therapeutic agents for nasal delivery or through the respiratory route, clear airway passages are a requirement for such treatments.

BromAc, a combination of bromelain and acetylcysteine, has shown both anticancer and mucolytic properties [26–28] and is currently undergoing Phase 2 clinical evaluation for the treatment of pseudomyxoma peritonei (PMP), where cancer cells secrete copious amounts of mucin in the peritoneal cavity, which eventually restricts nutritional intake resulting in death [29]. Since both agents in BromAc possess mucolytic properties, demonstrate synergy [27] and have antimicrobial properties [30,31], they may serve in solubilization of mucinous secretion whilst also acting as an anti-microbial agent. Owing to BromAc's strong mucolytic activity, with its hydrolytic action on peptide and glycosidic bonds, along with its disulfide reductive properties [32,33], it is envisaged that these two agents may solubilize dense and purulent sputa secreted in the airways in disease states. Preliminary study with BromAc on cystic fibrosis sputa has shown that it clearly disintegrates the sticky mucinous mass into a free-flowing solution in vitro (data attached in the Appendix A), and

hence, in the current study, we investigate the mucolytic effect of nebulized BromAc on two sputa models, since we envisage using this formulation in aerosol form for treatment. The two sputa models used in this study are artificial sputa (AS) as well as simulated sputa (SS) that is mucinous secretion from PMP patients which has been specially treated to represent thick airway sputa. Nebulized delivery of therapeutics is a convenient method for treating diseases of the airways [34].

## 2. Materials and Methods

### 2.1. Materials

For preparing artificial sputa, the following materials were purchased from Sigma Aldrich, Sydney, Australia: porcine mucin, salmon sperm DNA, potassium chloride, sodium chloride, TRISMA Base, TPTZ (2,4,6-Tripyridyl-s-triazine), and ferrous chloride. PMP mucin of soft grade was obtained from a clinical sample that had been assessed for its hardness index [35]. Additional materials included pipettes (0.5 mL), capillary tubes, endotracheal tubes size 9.0, and nebulizer equipment (InnoSpire Essence Nebulizer Compressor (PHILIPS, Amsterdam, The Netherlands), flow rate 7 L per minute, 10 psi).

### 2.2. Measurement of Size Distribution of Aerosols

The size distribution of the droplets emitted from the InnoSpire Essence Nebulizer device was measured on a laser diffractometer (Spraytec®, Malvern Panalytical, Malvern, UK) over 30 min. The Spraytec measures real-time, in situ particle size distribution of the aerosols passing through a laser beam. The aerosolized droplets were sized with an inhalation cell and at an acquisition frequency of 2.5 kHz. The outlet of aerosols was positioned 1 cm from the laser measurement zone to minimize evaporation during measurement. The raw data were processed to yield an averaged volumetric diameter distribution for a period in a given run. Particle size distributions were expressed as d10 (volume diameter under which 10% of the particles reside), d50 (volume median diameter), d90 (volume diameter under which 90% of the particles reside), and geometric standard deviation (GSD), which describes the polydispersity of the aerosols.

### 2.3. Artificial Sputa Preparation

Artificial sputum was prepared following protocol as detailed by Kirchner [36]. Briefly, 250 mg of porcine mucin, 200 mg of sperm DNA, 295 mg of diethylene triamine penta-acetic acid (DPTA), 25 mg of sodium chloride, 110 mg of potassium chloride, and 140 mg of Tris base were mixed in a volume of 30 mL of distilled water and pH adjusted to 7.0 using Tris base. The volume of the solution was then adjusted to 35 mL. Further dilutions were then carried out to adjust viscosity as desired.

### 2.4. Preparation of PMP Mucin as a Model of Sputa

Six grams of soft PMP mucin was homogenized using a shredder in phosphate buffer saline (PBS) (3.0 mL) with sonification and vortexed until a homogenous mixture was formed, with incubation at 37 °C to remove air bubbles, after which viscosity was adjusted (further dilutions with PBS). The pH was adjusted to 7.0 using either 1.0 M NaOH or 0.1 N hydrochloric acid.

### 2.5. Measurement of Apparent Viscosity

Apparent viscosity ($\gamma$) of sputum was measured using the capillary tube method as outlined in reference [37].

### 2.6. Measurement of Pipette Flow Time

Using a 0.5 mL glass pipette fixed at an angle of 60°, 0.5 mL of sample at 25 °C (ambient room temperature) was sucked up the pipette, and the time taken to empty 0.3 mL of the sample was timed in second.

Pipette flow time ($\varepsilon$) was calculated as follows:

$$\varepsilon \text{ (mL/s)} = 0.3 \text{ mL/time to empty 0.3 mL (s)}$$

### 2.7. Apparatus Set-Up

The nebulizer equipment was set up as shown in Figure 1, with the endotracheal tube containing the sputa sample immersed in a water bath at 37 °C. After each reagent (6.0 mL) was delivered over 25 min, the nebulizer receptacle for the reagent was washed along with the endotracheal tubes, before commencing the next treatment.

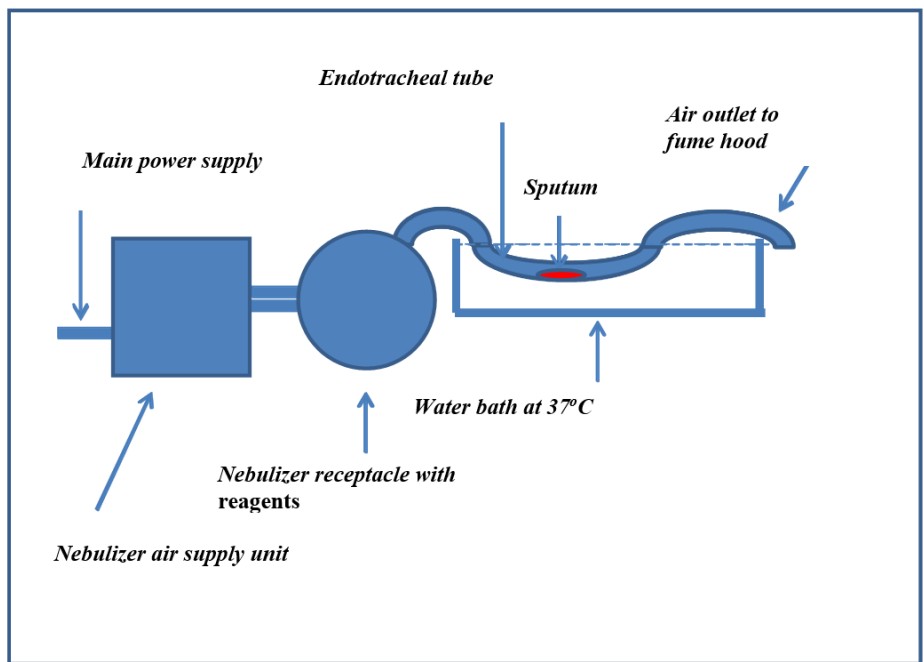

**Figure 1.** A schematic diagram showing the layout of equipment used in the experiment.

### 2.8. Treatment of Artificial Sputa (AS)

Artificial sputa were prepared as detailed earlier, and 1.5 mL of sputum was carefully emptied into an endotracheal tube (in triplicates). A volume of 6.0 mL of saline or other reagents in PBS (pH 7.0) in triplicates was aerosolized using a nebulizer and passed over the sputum samples in the endotracheal tubes kept at 37 °C in a water bath. The aerosol delivery time to empty 6.0 mL of reagent for each treatment was 25 min.

The apparent viscosity of sputa was measured before and after treatment using the capillary tube method, whilst the pipette flow time was also measured as described earlier using a 0.5 mL glass pipette. Samples were equilibrated to 25 °C before measurement.

### 2.9. Treatment of PMP Mucin Simulated Sputa (SS)

A similar procedure to the above was carried out to determine a comparative apparent viscosity and pipette flow time.

### 2.10. Measurement of Bromelain in Aerosolized Sputum Samples

Suitable dilutions such as 1/5 and 1/10 of nebulized samples in PBS were carried out with subsequent filtration (0.44 μm). To 250 μL of sample solution was added 250 μL of 1% azocaesin solution (prepared in distilled water). The samples were agitated at room temperature (25 °C) for 1 h, after which 1.5 mL of 1% trichloroacetic acid was added, vortexed and centrifuged at 2500 rpm. To 150 μL of supernatant in a microwell was added an equal quantity of 0.5 M sodium hydroxide solution, and the OD at 410 nm was read using a spectrometer (Shimadzu, Kyoto, Japan).

A standard curve for bromelain was generated following a similar procedure with bromelain dilutions ranging from 200 to 1.56 μg/mL (serially dilution).

### 2.11. Measurement of Acetylcysteine in Sputum Samples

Suitable dilution, such as 1/5 and 1/10 of aerosolized samples with filtration as above, was carried out. A 10 mM quantity of solution of TPTZ was prepared in distilled water (dH2O). Stock solution of NAC 10 mg/mL was prepared in PBS and pH adjusted to 7.0. Stock solution (10 mM) of Fe (III) was prepared in dH2O. The pH of all the reagents was adjusted to 7.0.

To 125 μL of TPTZ was added 125 μL of Fe (III) solution vortex, mixed and then followed by the addition of 100 μL of test solution. It was then vortexed and placed on a gentle shaker for 1 h at ambient room temperature (25 °C) until full color (blue) developed. To this was added 2.0 mL of $dH_2O$, vortex mixed, and the OD was measured using aliquots of 200 μL in triplicates in a 96-well plate. Blanks only contained 100 μL of dH2O. The OD at 593 nm was read using a UV spectrometer (Shimadzu, Kyoto, Japan). Suitable dilutions of acetylcysteine from 200 μg/mL down were prepared for the generation of the standard curve.

Calculation of D values (%) for both the dynamic viscosity ($\gamma$) and pipette flow ($\varepsilon$):

$$D = [(TREATED - UNTREATED)/UNTREATED] \times 100.$$

### 2.12. Determination of Combination Index (CI) of Bromelain and N-Acetylcysteine in Affecting Viscosity and Flow Speed

The combination index was determined using Chou and Tallaly's method as in reference [38]. Values below 0.99 were classified as synergy, >0.99–1.1 as additive, and above 1.1 as sub-additive.

### 2.13. Statistical Analysis

Data were reported as the mean $\pm$ SD. Qualitative variables were compared using Student's *t*-test. Test samples were compared with control after treatment for significance testing. Differences were considered statistically significant when $p < 0.05$.

## 3. Results

### 3.1. Size Distribution of Aerosols Emitted from the Jet Nebulizer InnoSpire

The jet nebulizer produced aerosols with volumetric median diameter smaller than five microns for all BromAc concentrations. The droplet size distribution was stable over the entire period of nebulization. Similar particle size distributions were observed for all formulations. The d50 of all formulations was smaller than 5 μm, while the d90 of all formulations was smaller than 10 μm. Sizes of more than 96% of particles of formulations were <10 μm in diameter. While sizes of more than 69% of particles of formulations were <5 μm in diameter (Table 1).

**Table 1.** Volumetric droplet diameter distributions of BromAc aerosols emitted from the jet nebulizer InnoSpire.

| ID | $d_{10}$ (μm) | $d_{50}$ (μm) | $d_{90}$ (μm) | GSD | <10 μm (%) | <5 μm (%) | Nebulisation Time (min, s) |
|---|---|---|---|---|---|---|---|
| | Bromelain 250 μg/mL + 20 mg/mL acetylcysteine | | | | | | |
| Mean | 1.6 | 3.6 | 7.8 | 1.7 | 96.3 | 69.4 | 30, 19 |
| Standard Deviation | 0.1 | 0.1 | 0.3 | 0 | 0.61 | 1.42 | 1.5, 10 |
| | Bromelain 500 μg/mL + 20 mg/mL acetylcysteine | | | | | | |
| Mean | 1.7 | 3.5 | 7.0 | 1.7 | 98.1 | 74.2 | 30, 26 |
| Standard Deviation | 0.1 | 0.1 | 0.3 | 0 | 0.46 | 2.00 | 0.6, 25 |
| | Bromelain 1000 μg/mL + 20 mg/mL acetylcysteine | | | | | | |
| Mean | 1.5 | 3.5 | 7.5 | 1.8 | 96.8 | 71.9 | 30, 32 |
| Standard Deviation | 0.1 | 0.2 | 0.3 | 0 | 0.40 | 2.51 | 1.2, 25 |

### 3.2. Treatment with N-Acetylcysteine (NAC)

Treatment with PBS had a very minor effect on viscosity (γ) of artificial sputum (AS); however, NAC at 10 and 20 mg/mL showed a marked decrease in γ, 6.0 and 9.8%, respectively. The pipette flow speed (ε) indicated that treatment with PBS affected it by 20%, indicating that hydration may play a substantial role on this parameter. Further, treatment with NAC 10 and 20 mg/mL also showed a corresponding increase in pipette flow (28 and 40%, respectively).

In the case of simulated sputa (SS), PBS treatment affected γ slightly (2%), while having almost an equal effect (16 and 17%) with treatment of NAC at the two concentrations (10 and 20 mg/mL). Further, there was also a corresponding increase in ε values. Thus, both sputa models were affected by aerosol treatment with NAC (Table 2 and Figure 2A–D).

**Table 2.** The effect of aerosolized N-Acetylcysteine on both artificial and simulated sputa using two parametric measurements such as viscosity and pipette flow time. NAC: N-Acetylcysteine. Data presented as mean ± SD. Test samples were compared with control after treatment for significance testing; *: $p < 0.05$; ↓ = decrease; ↑ = increase.

| Viscosity (γ) Measurements (Pa·s) | | | | | | |
|---|---|---|---|---|---|---|
| | Artificial Sputum (AS) | | | Simulated Sputum (SS) | | |
| Treatment | Before | After | D (%) | Before | After | D (%) |
| Control | | 22.383 ± 0.017 | 0.65 ↓ | | 28.550 ± 0.067 | 2 ↓ |
| NAC 10 mg/mL | 23.038 ± 0.217 | 21.667 ± 0.033 * | 6.0 ↓ | 29.133 ± 0.1 | 24.450 ± 0.017 * | 16 ↓ |
| NAC 20 mg/mL | | 20.783 ± 0.683 | 9.8 ↓ | | 24.217 ± 0.917 * | 17 ↓ |
| Pipette Flow Speed (ε) (mL/s) | | | | | | |
| | Artificial Sputum (AS) | | | Simulated Sputum (SS) | | |
| Treatment | Before | After | D (%) | Before | After | D (%) |
| Control | | 0.006 ± 0.001 | 20 ↑ | | 0.0153 ± 0.001 | 12 ↑ |
| NAC 10 mg/mL | 0.005 ± 0.001 | 0.0064 ± 0.003 | 28 ↑ | 0.0137 ± 0.005 | 0.0183 ± 0.001 * | 34 ↑ |
| NAC 20 mg/mL | | 0.007 ± 0.001 | 40 ↑ | | 0.02 ± 0.002 * | 46 ↑ |

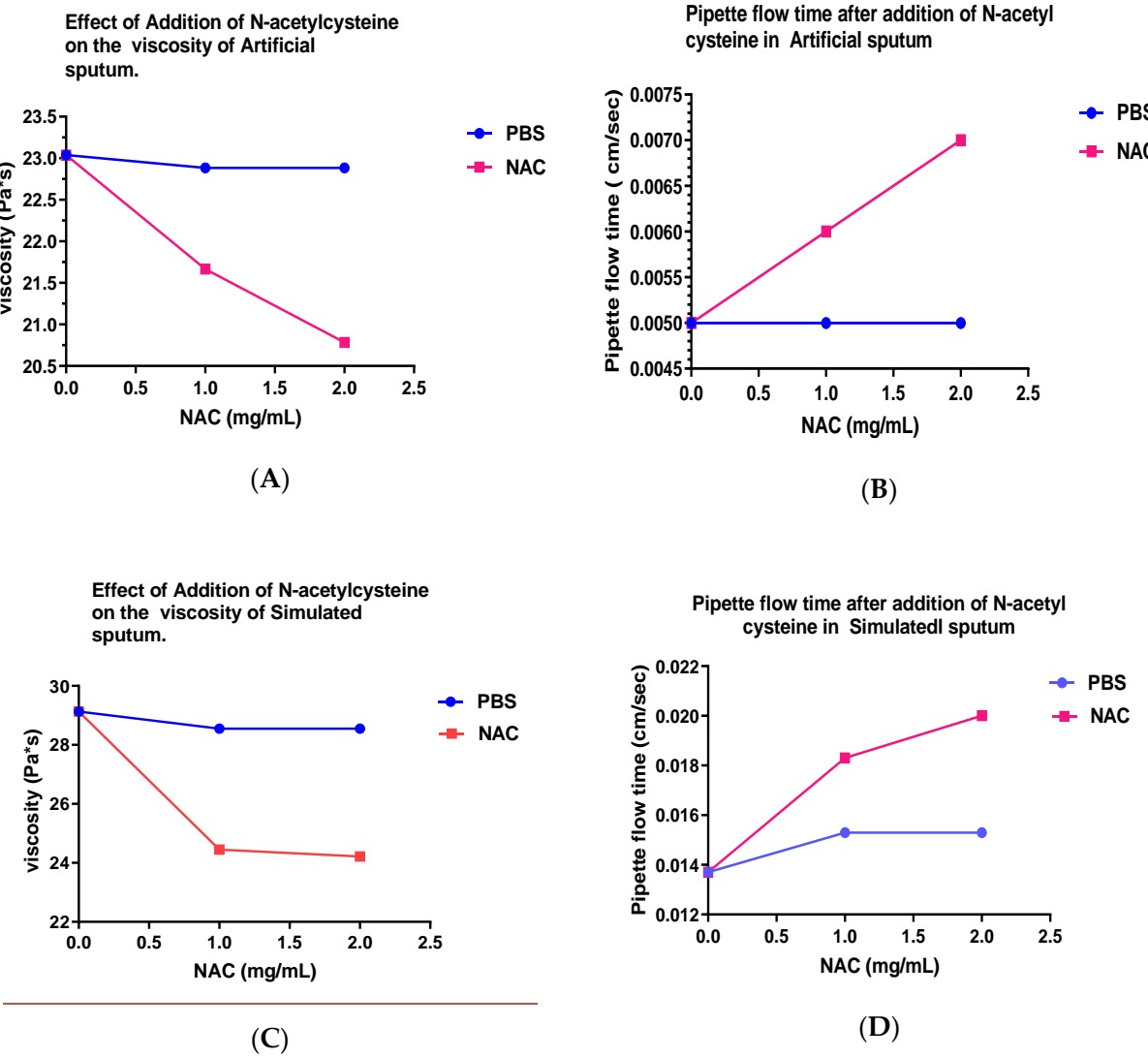

**Figure 2.** (**A**) In artificial sputa (AS), with the addition of N-Acetylcysteine, there was a noticeable concentration-dependent reduction of viscosity. (**B**) In AS, there was a linear increase in pipette sputum flow with the addition of increasing amounts of NAC. (**C**) There was a considerable drop in viscosity of simulated sputum (SS) with the addition of NAC 10 and 20 mg/mL, as compared to control. (**D**) The increase in pipette sputum flow after the addition of increasing amounts of NAC in SS model.

### 3.3. Treatment with Bromelain (BR)

The viscosity (γ) of both the sputa AS and SS were slightly affected by aerosol PBS, 0.65 and 2.0%, respectively; however, treatment with bromelain at 125 and 250 µg/mL indicated a noticeable drop in (γ) for both the sputa models, with much higher drops for AS. The effect at the two concentrations of bromelain in the SS model was almost the same. When the pipette flow speed (ε) was examined, aerosol PBS showed an effect on both sputa models: 20 and 12% in AS and SS, respectively; however, this effect was very high when treated with bromelain in both the models (Table 3 and Figure 3A–D)

**Table 3.** The effect of aerosolized bromelain (BR) on both artificial (AS) and simulated sputa (SS) using two parametric measurements, viscosity ($\gamma$) and pipette flow time ($\varepsilon$). Data presented as mean $\pm$ SD. Test samples were compared with control after treatment for significance testing; *: $p < 0.05$; $\downarrow$ = decrease; $\uparrow$ = increase.

| Viscosity ($\gamma$) Measurements (Pa·s) | | | | | | |
|---|---|---|---|---|---|---|
| | Artificial Sputum (AS) | | | Simulated Sputum (SS) | | |
| Treatment | Before | After | D (%) | Before | After | D (%) |
| Control | | 22.883 $\pm$ 0.017 | 0.65 $\downarrow$ | | 28.550 $\pm$ 0.067 | 2 $\downarrow$ |
| BR 125 µg/mL | 23.038 $\pm$ 0.217 | 16.867 $\pm$ 0.367 * | 27 $\downarrow$ | 29.133 $\pm$ 0.1 | 26.867 $\pm$ 0.067 * | 8.0 $\downarrow$ |
| BR 250 µg/mL | | 14.50 $\pm$ 0.050 * | 36 $\downarrow$ | | 26.667 $\pm$ 0.150 * | 8.5 $\downarrow$ |
| Pipette Flow Speed ($\varepsilon$) (mL/s) | | | | | | |
| | Artificial Sputum (AS) | | | Simulated Sputum (SS) | | |
| Treatment | Before | After | D (%) | Before | After | D (%) |
| Control | | 0.006 $\pm$ 0.001 | 20 $\uparrow$ | | 0.0153 $\pm$ 0.001 | 12 $\uparrow$ |
| BR 125 µg/mL | 0.005 $\pm$ 0.001 | 0.0218 $\pm$ 0.003 * | 336 $\uparrow$ | 0.0137 $\pm$ 0.005 | 0.046 $\pm$ 0.001 * | 243 $\uparrow$ |
| BR 250 µg/mL | | 0.0290 $\pm$ 0.001 * | 480 $\uparrow$ | | 0.073 $\pm$ 0.004 * | 443 $\uparrow$ |

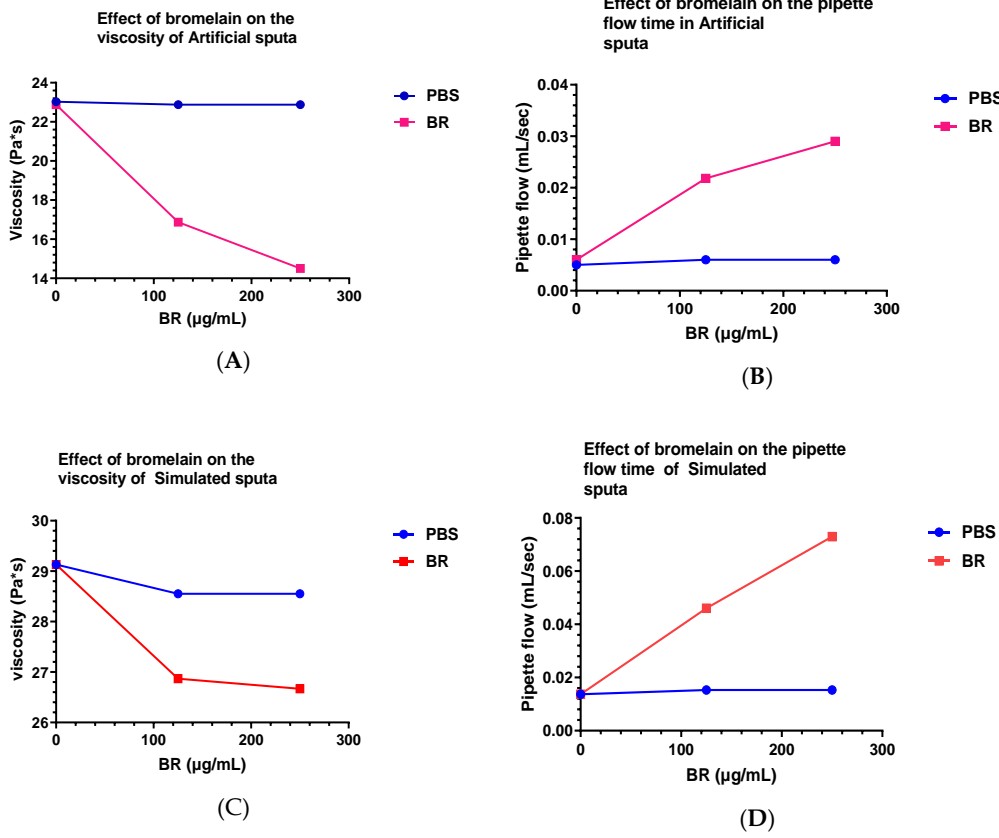

**Figure 3.** (**A**) In artificial sputum (AS), with the addition of bromelain (BR), there was a considerable reduction of viscosity in comparison to control (PBS). (**B**) The pipette flow in AS was considerably increased as compared to control (PBS) when treated with bromelain. (**C**) In simulated sputa (SS), the addition of bromelain shows a reduction in viscosity in comparison to control (PBS). (**D**) In SS, the pipette flow was highly increased compared to control (PBS) after treatment with bromelain.

### 3.4. Treatment with Bromelain and Acetylcysteine (BromAc)

Treatment with bromelain at the two concentrations (125 and 250 µg/mL) with NAC 20 mg/mL had a noticeable effect on ($\gamma$), with sufficient decrease in both sputa mod-

els. This was further indicated in (ε) values, with a very high increase in both models (Table 4 and Figure 4A–D).

**Table 4.** The effect of aerosolized BromAc (bromelain (BR) 125 or 250 μg/mL + NAC 20 mg/mL) on both artificial (AS) and simulated sputa (SS) using two parametric measurements such as viscosity (γ) and pipette flow time (ε). Data presented as mean ± SD. Test samples were compared with control after treatment for significance testing; * = *p* < 0.05; ↓ = decrease; ↑ = increase.

| Viscosity (γ) Measurements (Pa·s) | | | | | | |
|---|---|---|---|---|---|---|
| | Artificial Sputum (AS) | | | Simulated Sputum (SS) | | |
| Treatment | Before | After | D (%) | Before | After | D (%) |
| Control | | 22.883 ± 0.017 | 0.65 ↓ | | 28.550 ± 0.067 | 2 ↓ |
| BR 125 μg/mL + NAC 20 mg/mL | 23.038 ± 0.217 | 15.833 ± 0.017 * | 31 ↓ | 29.133 ± 0.1 | 23.6 ± 0.550 * | 19 ↓ |
| BR 250 μg/mL + NAC 20 mg/mL | | 13.333 ± 0.033 * | 42 ↓ | | 21.367 ± 0.450 * | 27 ↓ |
| Pipette Flow Speed (ε) (mL/s) | | | | | | |
| | Artificial Sputum (AS) | | | Simulated Sputum (SS) | | |
| Treatment | Before | After | D (%) | Before | After | D (%) |
| Control | | 0.006 ± 0.001 | 20 ↑ | | 0.0153 ± 0.001 | 12 ↑ |
| BR 125 μg/mL + NAC 20 mg/mL | 0.005 ± 0.001 | 0.033 ± 0.002 * | 556 ↑ | 0.0137 ± 0.005 | 0.061 ± 0.002 * | 343 ↑ |
| BR 250 μg/mL + NAC 20 mg/mL | | 0.035 ± 0.001 * | 600 ↑ | | 0.114 ± 0.002 * | 733 ↑ |

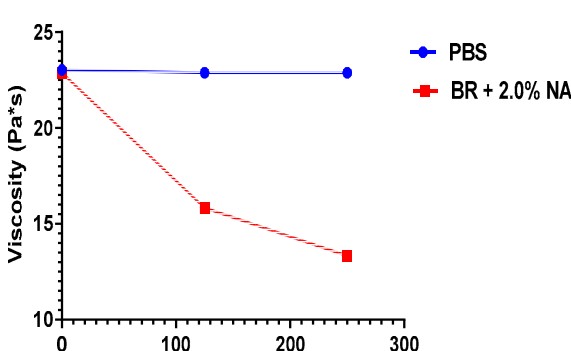

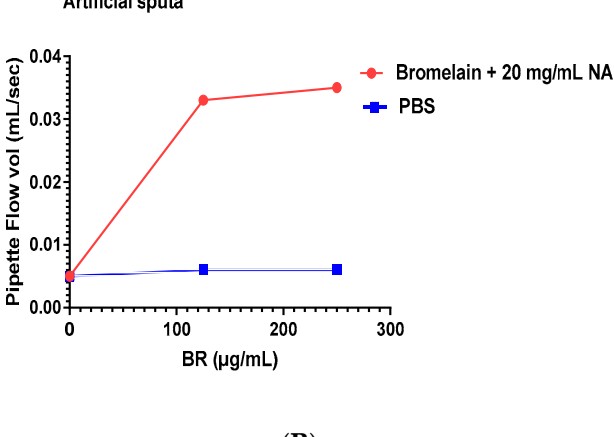

(A)  (B)

**Figure 4.** *Cont.*

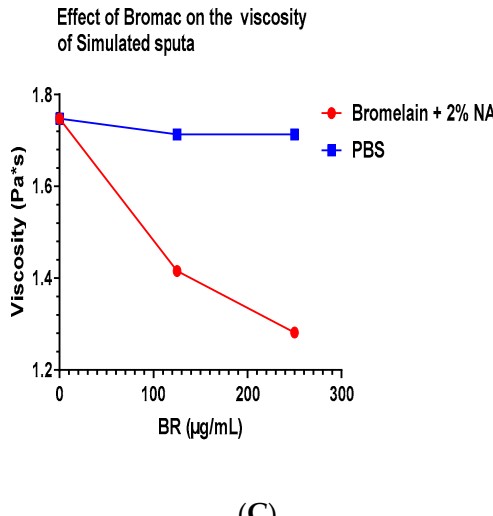

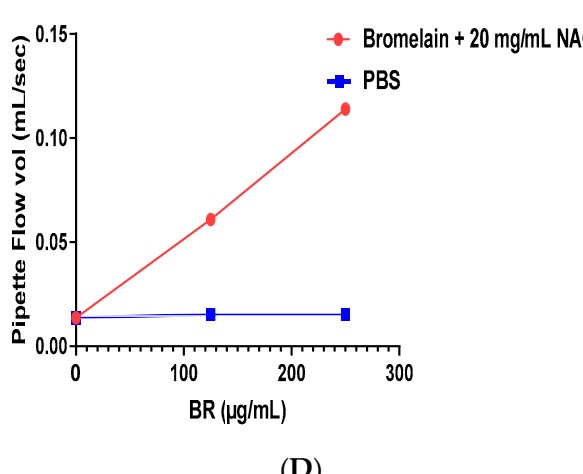

(**C**) (**D**)

**Figure 4.** (**A**) In artificial sputa (AS), there was a noticeable drop in viscosity after treatment with BromAc (bromelain 125 and 250 μg/mL + NAC 20 mg/mL). (**B**) In AS, the flow rate increased with the addition of BromAc. (**C**) In simulated sputa (SS,) there was a considerable drop in viscosity with the addition of 125 or 250 μg/mL bromelain + NAC 20 mg/mL. (**D**) In SS, there was almost a linear increase in flow rate (pipette emptying time) with the addition of increasing amounts of bromelain and 20 mg/mL NAC.

Bromelain at 125 μg/mL as an individual agent, showed a concentration of 30.58 μg/mL in the AS, and with 250 μg/mL it was 58.63 μg/mL, indicating that doubling the concentration almost doubled the sequestered bromelain. However, in the SS, there was very little difference between the two bromelain concentrations (57.91 vs. 61.64 μg/mL). NAC, as an individual agent, showed only a small difference in concentration between 10 and 20 mg/mL in AS, whilst slightly larger differences in SS models. When bromelain was delivered in NAC 20 mg/mL, bromelain concentration in the 125 μg/mL was half of that found in the 250 μg/mL bromelain in the AS model, whilst in the SS model, there was only 11% difference. Although NAC 20 mg/mL was delivered with either 125 or 250 μg/mL bromelain, NAC analysis indicated that in the AS model, the difference was small, with a slightly larger difference in the SS model (Table 5, Figure 5A–C).

**Table 5.** The concentration of either bromelain (BR) or acetylcysteine (NAC) in two sputa models (AS and SS) after passage of nebulized solution over the sputa with just bromelain, just NAC, or their combination, over 25 min.

| | Artificial Sputa (AS) | | Simulated Sputa (SS) | |
|---|---|---|---|---|
| Reagent | BR (μg/mL) | NAC (mg/mL) | BR (μg/mL) | NAC (mg/mL) |
| BR 125 μg/mL | 30.58 ± 0.211 | | 57.91 ± 0.562 | |
| BR 250 μg/mL | 58.63 ± 0.89 | | 61.64 ± 0.88 | |
| NAC 10 mg/mL | | 2.75 ± 0.143 | | 1.85 ± 0.310 |
| NAC 20 mg/mL | | 2.90 ± 0.210 | | 2.367 ± 0.133 |
| BR 125 μg/mL + NAC 20 mg/mL | 41.23 ± 1.212 | 2.41 ± 0.122 | 77.56 ± 3.12 | 1.62 ± 0.093 |
| BR 250 μg/mL + NAC 20 mg/mL | 79.15 ± 2.22 | 2.22 ± 0.184 | 88.21 ± 4.11 | 2.16 ± 0.132 |

Further, we compared the D values of both viscosity ($\gamma$) and flow speed ($\varepsilon$) between artificial and simulated sputa (Table 6 and Figure 6). They show that small changes in viscosity of the mucinous solution can result in large flow rates based on pipette emptying time. This may have a bearing on ciliary clearance, which needs further investigation.

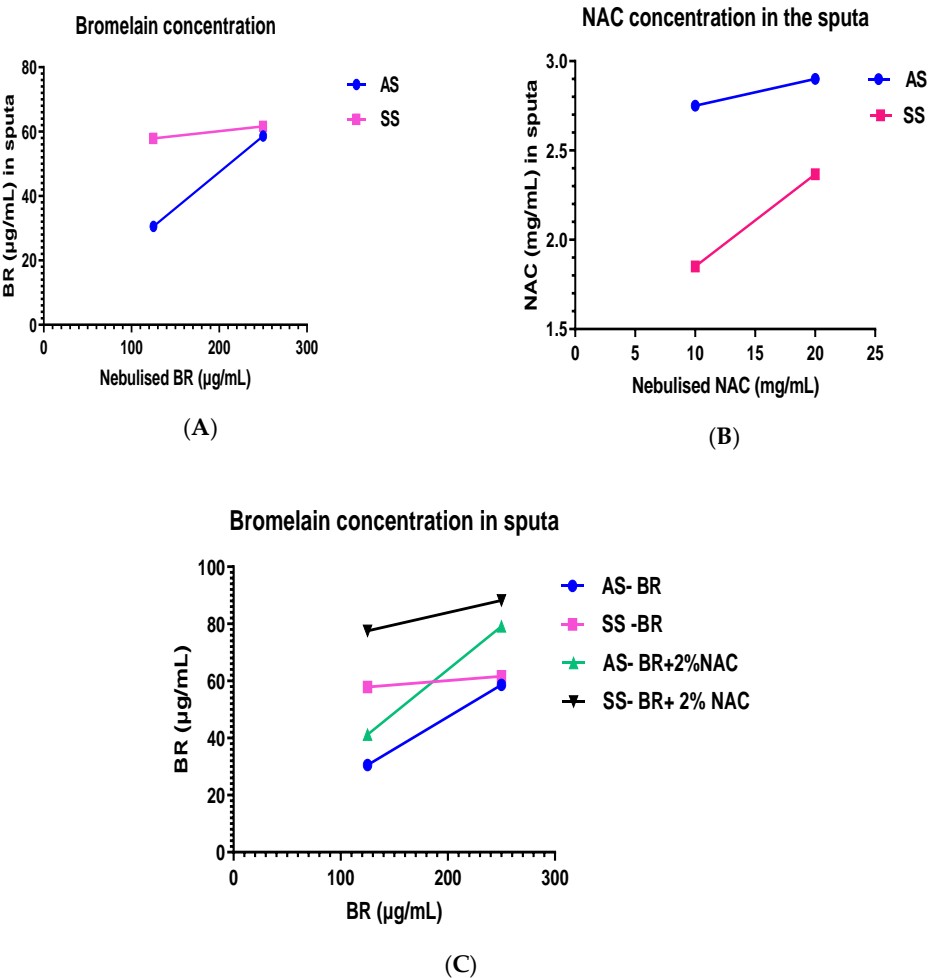

**Figure 5.** (**A**) The concentration of bromelain in the artificial sputa (AS) increases almost 2-fold with exposure to a two-fold increase in concentration of aerosolized bromelain (125 vs. 250 µg/mL). However, the difference was relatively small for the SS, simulated sputa, 58 vs. 62 µg/mL. (**B**) The difference between the low (10 mg/mL) and the high (20 mg/mL) NAC in the artificial sputa (AS) was relatively small (difference of 0.15 mg/mL) when exposed to aerosolized NAC. However, in the simulated sputa (SS), the difference was relatively larger (1.85 vs. 2.37 mg/mL), about a 28% increase. (**C**) Comparative levels of bromelain in artificial and simulated sputa before and after addition of 20 mg/mL NAC to the aerosolized solution. The concentration of bromelain is relatively higher in both sputa models in the presence of NAC 20 mg/mL.

**Table 6.** Comparison of viscosity ($\gamma$) and flow speed ($\varepsilon$) between the two sputa models when treated with bromelain 125 or 250 µg/mL and in combination with NAC 20 mg/mL.

| | D Value (%) | | | |
| --- | --- | --- | --- | --- |
| | Artificial Sputa (AS) | | Simulated Sputa (SS) | |
| Treatment | Dynamic Viscosity ($\gamma$) | Flow Speed ($\varepsilon$) | Dynamic Viscosity ($\gamma$) | Flow Speed ($\varepsilon$) |
| 125 µg/mL BR | 27 | 336 | 8.0 | 243 |
| 250 µg/mL BR | 36 | 480 | 8.2 | 443 |
| 125 µg/mL BR + NAC 20 mg/mL | 31 | 556 | 19 | 343 |
| 250 µg/mL BR + NAC 20 mg/mL | 42 | 600 | 27 | 733 |

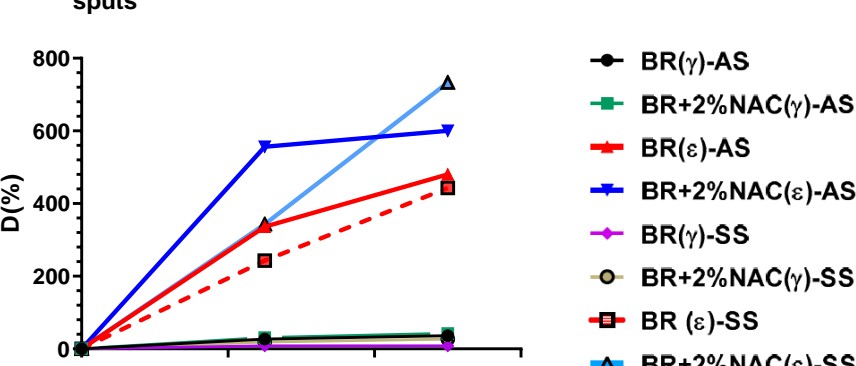

**Figure 6.** The relative differences in enhancement D (%) of the two parameters measured such as viscosity (**γ**) and flow speed ε in the artificial sputum (AS) and simulated sputum (SS), when treated with either bromelain (BR) 125 or 250 μg/mL alone or in combination with NAC 20 mg/mL (BromAc). Viscosity changes in the different groups are amplified in flow speed, showing the effect of viscosity on the latter. Treatment with BromAc has a much higher effect on flow speed compared to treatment with bromelain alone. D = [(Treated − Untreated)/Untreated] × 100.

Although with the addition of NAC 20 mg/mL to bromelain 125 and 250 μg/mL in artificial sputum, the decrease in viscosity was only 4–6 % compared to bromelain alone (from 31–42% to 27–36%, respectively), there was tremendous increase in flow speed ($\varepsilon$), indicating that small decreases (changes) in viscosity may affect clearance of the fluid by ciliary motion. The high BromAc concentration group indicates almost twice the reduction of pipette emptying time compared to the low BromAc group. This does not correlate with their respective viscosity ($\gamma$), which may indicate that high glycosylated mucin may affect the viscosity to a significant level. In simulated sputa, with the addition of NAC 20 mg/mL to bromelain 125 and 250 μg/mL, the decrease in viscosity was 19 and 27% compared to 8 and 8.2%, respectively. Further, a similar trend in flow speed ($\varepsilon$) was seen in both sputa.

The viscosity measurements indicate that when 125 μg/mL bromelain + 20.0 mg/mL NAC was reacted with both sputa models, the combination index was sub-additive, whilst the addition of 250 μg/mL bromelain + 20 mg/mL NAC indicated an additive interaction for artificial sputa with synergy in the simulated sputa, indicating the variability of the composition of the two sputa models. On the contrary, the flow speed indicated complete synergy for all the additions (Table 7).

**Table 7.** The combination index when NAC (2.0%) is combined with either bromelain at 125 or 250 μg/mL (BromAc) in both viscosity and flow speed measurement as determined by the Chou and Talalay method for artificial and simulated sputa.

| | | Viscosity | | | | Flow Speed | | | |
|---|---|---|---|---|---|---|---|---|---|
| | | Artificial Sputa | | Simulated Sputa | | Artificial Sputa | | Simulated Sputa | |
| BR (μg/mL) | NAC (mg/mL) | CI | Effect | CI | Effect | CI | Effect | CI | Effect |
| 125.00 | 20.0 | 1.19 | Sub-additive | 1.31 | Sub-additive | 0.672 | Synergy | 0.846 | Synergy |
| 250.00 | 20.0 | 1.1 | Additive | 0.94 | Synergy | 0.866 | Synergy | 0.666 | Synergy |

## 4. Discussion

Before we began exploring the effect of BromAc on the rheology of sputum, we measured the size of aerosolized BromAc particles to ensure that BromAc particles are

suitable for airway delivery. It has been shown that particles with sizes between 1 and 5 μm are deposited in secondary bronchi and bronchioles, and particles with sizes between 5 and 10 μm are impacted in primary bronchi [39,40]. Here, our data showed that 96% of particles of BromAc formulations have a size <10 μm in diameter, and 69% of particles of BromAc formulations have a particle size <5 μm in diameter. Hence, the tested BromAc formulations were suitable for aerosol delivery to primary and secondary bronchi as well as bronchioles.

Since clearance of airway secretion is mainly dependent on its rheological parameters, we decided to measure the viscosity (γ) and the flow speed (ε) of the sputa before and after treatment with BromAc using two model sputa: artificial sputa (AS) and simulated sputa (SS), specially formulated to represent thick and static sputa. However, since two different agents (bromelain and acetylcysteine) were incorporated into the formulation (BromAc), the effect of individual agents were first investigated with subsequent studies on their combination. The differences for both γ and ε (pre-treated as opposed to treated) were calculated as a percentage denoted by D. Additionally, we also investigated the sequestration of bromelain and NAC in the sputa before and after aerosol delivery for individual agents, as well as their combinations, since we wanted to correlate the concentrations of the agents within the sputa with the changes in rheological properties. Treatment of artificial sputa (AS) with aerosolized PBS indicated a minute drop in γ (0.65%) that may be mainly due to hydration, whilst in both the NAC 10 and 20 mg/mL, the reduction was by 6.0 and 10%, respectively. NAC is a well-known antioxidant, and hence the reduction of disulfide linkages found in the protein and mucin components may be responsible for this change in viscosity [41,42]. In comparison, simulated sputa (SS) with PBS treatment showed a 2.0% drop in γ, with substantial effect with NAC 10 and 20 mg/mL (γ = 16 and 17%, respectively). This difference between both sputa may be due to their differences in composition and variability of the constituents. The SS is mainly composed of mucinous mass that is heavily glycosylated with cellular debris and other components that may contain abundant disulfide linkages [43], and hence highly prone to the reductive action of NAC. Since the action of NAC was substantial in simulated sputa (SS), it may indicate that the percentage of disulfide linkages within the matrix may be much higher compared to artificial sputa (AS) (at NAC 20 mg/mL, γ was 17 and 10%, respectively). Statistical analysis showed significant differences in both sputa models (viscosities and flow speed) when the combination of bromelain and NAC was compared to individual agents, except for simulated sputa with the addition of NAC as a single agent (Appendix Table A1).

The effect on flow speed ε was substantial (20% increase) with PBS treatment, indicating that hydration alone may have considerable effect on the viscoelastic property of sputum in AS, which agrees with other researchers on the rheology of sputa [5,44]. Further, at NAC 10 and 20 mg/mL, the effect was 28 and 40%, respectively. In comparison, the SS displayed a substantial increase in flow speed ε with only PBS (12%), whilst both the NAC 10 and 20 mg/mL had a much higher impact (34 and 46% increase). The differences between the two sputa models in flow speed may be mainly due to their differences in composition. The percentage of disulfide bonds in the SS model may be much higher compared to the AS model, and hence, their reduction by NAC shows a considerable effect on this parameter.

Treatment with bromelain showed a marked drop in γ in both treatment groups (0.125 and 250 μg/mL): 27 and 36%, respectively, in the AS model, with substantial effect on ε (336 and 480%, respectively), indicating the impact of viscosity on the flow speed of sputa and ciliary clearance [45]. In the case of the SS model, again there was drop in γ, 8.0 and 8.2% for the two concentrations of bromelain (125 and 250 μg/mL, respectively), with almost no difference. This similarity in γ may be due to the high glycosylation and high mucin content present with the sputa. However, the effect of bromelain on γ amplified the effect on the ε values, as indicated (243 and 443%). Hence, bromelain seems to effect both parameters monitored, showing that its hydrolytic properties on proteins and glycoproteins affect the rheological properties of both sputa models. The variation in efficacy between the

two models may be attributed to their differences in composition. In comparison to NAC, bromelain has a much greater effect on the two rheological parameters monitored in AS; although less on $\gamma$, it had much higher impact on $\varepsilon$ values for the SS model, indicating that the enzymic reactions on these models' sputa show a much higher activity with greater depolymerization effect that affected the parameters monitored. Of note, in the SS model, the effect on $\gamma$ values was higher in the NAC treatment compared to bromelain, which may indicate the high disulfide content in the sample sputa. This needs further verification; however, studies have shown the depolymerizing effect of NAC through the reduction of these vital linkages that dictate the molecular geometry of proteins [42,46].

When the sputa were treated with BromAc (bromelain 125 and 250 μg/mL + NAC 20 mg/mL), $\gamma$ was considerably affected, as shown by the percentage differences between untreated and treated sputa in AS for both the 125 and 250 μg/mL bromelain, with values of 36 and 42%. Further, this effect was amplified in the $\varepsilon$ parameter with corresponding 556 and 600%. In the case of SS, the $\gamma$ parameter between 125 and 250 μg/mL bromelain showed 19 and 27%, with a substantial difference in $\varepsilon$ values, (343 and 733%, respectively). Hence, the effect on viscosity was really amplified in $\varepsilon$, with a considerable rise, because of treatment with bromelain plus NAC 20 mg/mL. These variations between the two models may be mainly due to their variability in composition, which needs further studies.

The concentration of bromelain in the sputa after treatment indicated that with bromelain alone, AS showed double the concentration when treated with 250 μg/mL compared to 125 μg/mL bromelain (58.63 vs. 30.58 μg/mL, respectively), with some correlation to their observed effect on $\gamma$. On the contrary, the concentration of bromelain in SS was almost similar in both bromelain groups (57.91 vs. 61.64 μg/mL), showing that bromelain may have perhaps accumulated to saturation in the models over 25 min aerosol delivery. The differences between the two sputa may also be related to their heterogenous composition. In the case of NAC, the concentration measured in the AS sputa seems to correlate with their activity as measured by dynamic viscosity, and likewise for the SS model. A similar correlation was seen with $\varepsilon$.

Finally, the analysis for bromelain in the BromAc (bromelain + NAC 20 mg/mL) indicated that there was a corresponding double concentration in the 250 μg/mL bromelain as opposed to 125 μg/mL bromelain (79 vs. 41 μg/mL bromelain) in the AS model, with correlation to their activity. This was not the case with the SS model that showed only about 11 μg/mL less in the 125 μg/mL group (78 vs. 88 μg/mL), indicating that near-saturation of bromelain may have taken place. In the BromAc group, the NAC sequestered was almost similar in both the high and low bromelain group, since NAC 20 mg/mL was delivered in both groups in the AS model. However, there was a difference in the SS model with the 125 μg/mL bromelain having less NAC concentration (1.62 vs. 2.16 mg/mL). Hence, these fluctuations may be partly due to differences in composition.

Viscosity ($\gamma$) and flow speed ($\varepsilon$) on a comparable basis indicate that both sputa models were affected. The $\gamma$ in SS was affected less by bromelain as compared to AS, and this may be due to their differences in composition, the former having higher protein and glycosidic linkages compared to the latter. However, when treated in combination with NAC 20 mg/mL as compared to bromelain alone, the effect on $\gamma$ was higher in SS, indicating that NAC may be playing a crucial role in reduction of disulfide bridges in the sample, affecting the rheology of the sputum. On the other hand, when examining, $\varepsilon$, both sputa were well affected. There is a marked difference in $\varepsilon$ between bromelain and that in combination with NAC, indicating the importance of NAC in depolymerizing the sputa. Importantly, these results further emphasize the high impact of these agents (bromelain, NAC, BromAc) on $\gamma$ and $\varepsilon$, such that any slight increase in the former magnifies the latter. This finding is important, particularly in developing formulas for improving the flow and clearance of sputum from the lungs.

Determination of combination index (CI) showed that the viscosity of sputa for both the sputa models were sub-additive with the addition of 125 μg/mL bromelain + 20.0 mg/mL NAC, whilst the addition of 250 μg/mL bromelain + 20 mg/mL NAC showed sub-

additivity for artificial sputa and synergy for the simulated sputa. These differences may be mainly due to their differences in chemical composition. However, when CI was determined using the flow speed, synergy was seen for both bromelain concentrations with 20 mg/mL NAC. This observation may indicate that the flow speed of sputa is affected more by slight variations in viscosities.

The effect of the individual agents and their combinations showed clearly that they have substantial effect on both sputa models, with variable effects on the two parameters measured owing to their heterogenous composition. The artificial sputa were laboratory-formulated, containing mainly porcine mucin, DNA, and a number of other additives, which did not include hyaluronic acid found in cystic fibrosis and COVID-19 sputa. On the other hand, the simulated sputa (SS) contained a heavy load of mucinous mass, which is substantially glycosylated (-O- linkages) and is prone to enzymic action of bromelain, whilst having a high load of cellular materials, lipids, sialic acid, etc. [43]. However, the observed rheological effects in these two model sputa may be sufficient to indicate that aerosol delivery of BromAc will have considerable effect on patient sputum from either CF or COVID-19. Recent studies have indicated that cystic fibrosis and COVID-19 airway secretions are heavily loaded with double-stranded DNA (>600 μg/mL) and hyaluronic acid (7.0 μg/mL) [17]. In comparison, the artificial sputa (AS) used in the present study contained about 5714 μg/mL DNA and 7142 μg/mL mucin, although we did not add hyaluronic acid. Hence, the variability in COVID-19 and CF sputum should not present any barrier to BromAc, since bromelain will hydrolyze the β-1-4 glycosidic linkages found in hyaluronic acid, since it has been widely used in hydrolysis of chitosan, a complex heavily glycosylated glycoprotein [32]. Importantly, CF and COVID-19 sputa share a very common chemical identity, since they have similar composition [17]. Further, the SS model used in this study is heavily glycosylated with abundant glycosidic bonds, and the mucolytic efficacy of BromAc in these sputa is a clear indication of its efficacy on chemical compounds containing the β1-4 glycosidic linkages and disulphide bonds that are found in hyaluronic acid (Figure 7). Although the current study is very promising in the development of BromAc for application in CF and COVID-19, further investigation using patient sputum with aerosol BromAc is warranted.

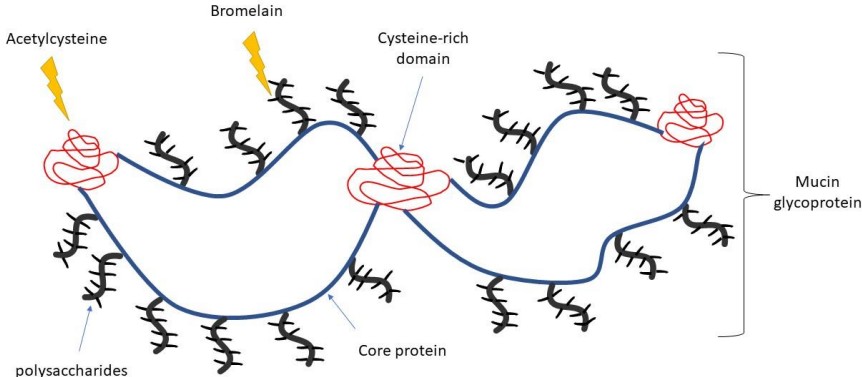

**Figure 7.** An illustration showing bromelain hydrolysis of glycosidic linkages that are abundant in mucin, whilst N-acetylcysteine reduces the di-sulfide linkages that are abundant cystine-rich domains in mucinous mass.

**Author Contributions:** Conceptualization, D.L.M., K.P., A.H.M., and J.A.; methodology, D.L.M., K.P., A.H.M. and J.A.; formal analysis, K.P. and A.H.M.; investigation, K.P. and A.H.M.; writing—original draft preparation, K.P.; writing—review and editing, D.L.M., K.P., A.H.M. and J.A.; visualization, K.P. and A.H.M.; supervision, D.L.M.; project administration, J.A.; funding acquisition, D.L.M. All authors have read and agreed to the published version of the manuscript.

**Funding:** This research is funded by Mucpharm Pty Ltd., Australia.

**Institutional Review Board Statement:** Not applicable.

**Informed Consent Statement:** Not applicable.

**Data Availability Statement:** The datasets generated during and/or analyzed during the current study are available from the corresponding author on reasonable request.

**Conflicts of Interest:** D.L.M. is the co-inventor and assignee of the license for this study and director of the spin-off sponsor company, Mucpharm Pty Ltd. K.P., A.H.M., and J.A. are employees of Mucpharm Pty Ltd.

## Appendix A

**Table A1.** Data presented as mean $\pm$ SD. Test samples were compared with control after treatment for significance testing; * = $p < 0.05$.

| Viscosity ($\gamma$) Measurements (Pa·s) | | |
|---|---|---|
| Treatment | Artificial Sputum (AS) | Simulated Sputum (SS) |
| BR 125 µg/mL + NAC 20 mg/mL | 15.833 $\pm$ 0.017 | 23.6 $\pm$ 0.550 |
| BR 125 µg/mL | 16.867 $\pm$ 0.367 * | 26.867 $\pm$ 0.067 * |
| NAC 20 mg/mL | 20.783 $\pm$ 0.683 * | 24.217 $\pm$ 0.917 |
| BR 250 µg/mL + NAC 20 mg/mL | 13.333 $\pm$ 0.033 | 21.367 $\pm$ 0.450 |
| BR 250 µg/mL | 14.50 $\pm$ 0.050 * | 26.667 $\pm$ 0.150* |
| NAC 20 mg/mL | 20.783 $\pm$ 0.683 * | 24.217 $\pm$ 0.917* |
| Pipette Flow Speed ($\varepsilon$) (mL/s) | | |
| Treatment | Artificial Sputum (AS) | Simulated Sputum (SS) |
| BR 125 µg/mL + NAC 20 mg/mL | 0.033 $\pm$ 0.002 | 0.061 $\pm$ 0.002 |
| BR 125 µg/mL | 0.0218 $\pm$ 0.003 * | 0.046 $\pm$ 0.001 * |
| NAC 20 mg/mL | 0.007 $\pm$ 0.001 * | 0.02 $\pm$ 0.002 * |
| BR 250 µg/mL + NAC 20 mg/mL | 0.035 $\pm$ 0.001 | 0.114 $\pm$ 0.002 |
| BR 250 µg/mL | 0.0290 $\pm$ 0.001 * | 0.073 $\pm$ 0.004 * |
| NAC 20 mg/mL | 0.007 $\pm$ 0.001 * | 0.02 $\pm$ 0.002 * |

Appendix Table A1 showed that statistical significance was observed in viscosity between BR + NAC (combination) and individual agent BR or NAC treatment in artificial sputum. On the other hand, in simulated sputum, bromelain showed significance in comparison to dual agents, whilst NAC did not. Bromelain is a proteolytic agent that hydrolyzes the peptide and glycosidic linkages, whilst NAC is a reducing agent that disrupts the disulfide linkages. The distribution or total content of disulfide linkages in the two models vary; the simulated sputa may contain a much higher content, and hence NAC as a single agent did not show much of a difference in viscosity changes. However, with bromelain in combination, significance was observed.

Sputum flow speed seems to be affected greatly by small changes in viscosities of the sputum, and hence statistical significance was shown at all the different additions.

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
