# Peer review of "Effect of Nebulized BromAc on Rheology of Artificial Sputum: Relevance to Muco-Obstructive Respiratory Diseases"

_arm, doi:10.3390/arm91020013_

Round 1

Reviewer 1 Report

The subject of the paper is very interesting and crucial to establish well designed treatment to muco – obstructive respiratory diseases. The rheology of the biological fluids such as mucus, sputum, blood, saliva, sperm etc. are very difficult to describe and far away comparing to well-known Newtonian fluids like water and air. Most of biological fluids are not Newtonian pseudo plastic or viscoelastic shear thinning fluids. Sputum is the example of such a fluid. 

During lecture of the paper, I have found some issues which should be explained before publication: 

1)    In all paper the authors use dynamic viscosity. In my opinion it is a mistake which should be corrected. Using name dynamic viscosity is proper only for Newtonian fluids. Non-Newtonian fluids like sputum (it is shear thinning fluid) have viscosity depended on the shear it means that for different flow conditions value of the apparent viscosity (it is correct name) is changing. Taking this into account I am afraid that the measuring apparent viscosity using capillary viscosimeter may give unconclusive results. The rheological experiments should be performed using rheometer ( e.g. Anthon Parr ) it will give more precise information about sputum rheology. 

1.1)         Viscosity is measured in Pa*s not in Pa ( p. 1 line 41). In tables authors use the cSc units which is not Si unit and should be recalculated into Pa*s

2)    The whole concept of the measurements looks clumsy … Why the authors decided to work with nebulizer and endotracheal tube?  In this set up there are a lot of problems with dose calibration. Did authors checked the droplet size distribution emitted from the nebulizer? Was it the same for all measurements? The dose of the drug administrated using inhalation strongly depend on the droplets diameter which depend on the viscosity of the fluid which is nebulized. In the set up presented in Figure 1 and flow of aerosol equal to 7 dm3/min I am pretty sure that the droplets deposited on the ET walls and run down the wall to the sample. How was the sample extracted from the tube when the droplets were deposited on the walls? In my opinion it would be much better and easer to simply introduce the drugs into the sputum sample using pipette. The nebulizer is redundant.

3)    The measurement of Bromelain and Acetylcysteine in sputum sample should be more precisely explained. The problem with spectrophotometry is that is not selective method. The presence of mucin affects the signal.  It should be proven that presence of mucin did not affect the results.

Author Response

Dear reviewer,

We thank you for reviewing our manuscript and for your valuable comments. we've highlighted changes in the new manuscript. Here is our response to your comments:

1)    In all paper the authors use dynamic viscosity. In my opinion it is a mistake which should be corrected. Using name dynamic viscosity is proper only for Newtonian fluids. Non-Newtonian fluids like sputum (it is shear thinning fluid) have viscosity depended on the shear it means that for different flow conditions value of the apparent viscosity (it is correct name) is changing. Taking this into account I am afraid that the measuring apparent viscosity using capillary viscosimeter may give unconclusive results. The rheological experiments should be performed using rheometer ( e.g. Anthon Parr ) it will give more precise information about sputum rheology.

Response: We have changed the term Dynamic viscosity to apparent viscosity as suggested. The measurement of viscosity using the capillary tube method is used as a convenient and simple method for just the comparison of viscosities before and after treatment. Although it may not be very precise, the measurement is solely used for just comparison in order to establish whether there is a difference.

1.1)         Viscosity is measured in Pa*s not in Pa ( p. 1 line 41). In tables authors use the cSc units which is not Si unit and should be recalculated into Pa*s

Response: The units of measurement have been changed to Pa’s as recommended.

2)    The whole concept of the measurements looks clumsy … Why the authors decided to work with nebulizer and endotracheal tube?  In this set up there are a lot of problems with dose calibration. Did authors checked the droplet size distribution emitted from the nebulizer? Was it the same for all measurements? The dose of the drug administrated using inhalation strongly depend on the droplets diameter which depend on the viscosity of the fluid which is nebulized. In the set up presented in Figure 1 and flow of aerosol equal to 7 dm3/min I am pretty sure that the droplets deposited on the ET walls and run down the wall to the sample. How was the sample extracted from the tube when the droplets were deposited on the walls? In my opinion it would be much better and easer to simply introduce the drugs into the sputum sample using pipette. The nebulizer is redundant.

Response:  We have determined the aerodynamic diameter of nebulizing reagents and they are mainly below 5.0 um since we used similar reagents for animal work by nebulization.

Although some of the aerosol reagents may have been deposited along the sides of the tube, they would have coalesced to form droplets that would have entered the sputum since the tube was curved creating a gradient for the droplets to enter the sputum. Further, the treatment is uniform and any error incurred would be uniform in all the treatments.

3)    The measurement of Bromelain and Acetylcysteine in sputum sample should be more precisely explained. The problem with spectrophotometry is that is not selective method. The presence of mucin affects the signal.  It should be proven that presence of mucin did not affect the results.

Response: Both the measurement of bromelain (azocaesin assay) and the N-acetylcysteine assay depends on specific properties of the compounds, bromelain depends on its proteolytic activity whilst N-acetylcysteine on its antioxidant potential. Mucin as such does not possess any proteolytic activity nor does it have any antioxidant properties.

We have included a more precise explanation of the assays.

Reviewer 2 Report

This research aims at understanding whether the combination of N-acetylcysteine and bromelain affects the rheological properties of sputum in a more efficacious way as compared to single agents.

To this end,  rheologic parameters, i.e. dynamic viscosity and sputum flow, of two different sputa models (AS and SS) were studied upon  aerosol N-acetylcysteine, bromelain, or their combination (BromAc).

The topic is relevant in the study of efficacious preparations of mucolytics that may ameliorate the clinical condition of patients with different morbidities, i.e. cystic fibrosis, COPD and COVID-19.

The study of this combination with aerosol was not assessed before.

The improvements are concerning aerosolization, which is the less invasive and more convenient method to concentrate drugs in the respiratory tract, improving their therapeutic index.

It is clear from their data that: i) the two sputa behave differently when treated with mucolytics, being different in composition; ii) the alterations of sputum flow are enhanced in comparison with dynamic viscosity changes.

My main criticism is that they have not used real-life sputum samples from CF or COPD or COVID-19 patients. A pool of sputum samples would be more appropriate. Said this, the study will be more valuable with more experimental samples including adding hyaluronic acid to AS samples in order to be closer to CF sputum composition. In alternative, they can use CF sputum samples.

Moreover, is the effect of the combination of the two agents an additive or a synergistic one?

Finally, no statistical analysis is presented throughout the paper and it should be added as Materials and Methods and Results.

I would add a cartoon displaying the mechanism of action of BromAc on the sputum rheology.

Author Response

Dear reviewer,

We thank you for reviewing our manuscript and for your valuable comments. we've highlighted changes in the new manuscript. Here is our response to your comments:

1) My main criticism is that they have not used real-life sputum samples from CF or COPD or COVID-19 patients. A pool of sputum samples would be more appropriate. Said this, the study will be more valuable with more experimental samples including adding hyaluronic acid to AS samples in order to be closer to CF sputum composition. In the alternative, they can use CF sputum samples.

Response: We have conducted preliminary work using sputa from cystic fibrosis patients. Although this would be more precise and more applicable to the live situation, the collection of sputa from patients is problematic. Hence, we decided to use laboratory-made sputa. However, the results from the patient were positive and we have included them in the paper as a supplement.

2) Moreover, is the effect of the combination of the two agents an additive or a synergistic one?

Response: We have included a new table showing the additive and synergistic effects of BromAc formulations (Table 7).

3) Finally, no statistical analysis is presented throughout the paper and it should be added as Materials and Methods and Results.

Response: There is a clear difference between untreated and treated samples. Hence, we feel that statistical analysis will not add more value in this case.

4) I would add a cartoon displaying the mechanism of action of BromAc on the sputum rheology.

Response: We have included an illustration of BromAc's action at the end of the discussion.

Round 2

Reviewer 1 Report

Manuscript in revised version is acceptable for publication in the journal but still some issues must be corrected. First of all the viscosity unit is Pascal (Newton divided by meter squared) multiplied by second. [Pa * s ] . On page 2 line 45 there is only Pa again. The same unit should be plased in Table, 2,3 and 4.    

 Authors wrote: 

"Although some of the aerosol reagents may have been deposited along the sides of the tube, they would have coalesced to form droplets that would have entered the sputum since the tube was curved creating a gradient for the droplets to enter the sputum. Further, the treatment is uniform, and any error incurred would be uniform in all the treatments.

In my opinion, the dose of API introduced into the sputum with your kit may vary, especially due to the process of deposition of droplets on the walls. The error may not be critical and may be omitted from the calculations. Despite this, I still don't understand why it was decided to use nebulization in the endotracheal tube system instead of pipetting the drug into the sputum. Nebulization was not the focus of the research...

Author Response

Dear reviewer,

We thank you for your revision of our manuscript and for your comments. we've highlighted changes in the new manuscript. Here is our response to your comments:

1) Manuscript in revised version is acceptable for publication in the journal but still some issues must be corrected. First of all the viscosity unit is Pascal (Newton divided by meter squared) multiplied by second. [Pa * s ] . On page 2 line 45 there is only Pa again. The same unit should be plased in Table, 2,3 and 4.

Response: This has been changed to Pascals.

2) Authors wrote: "Although some of the aerosol reagents may have been deposited along the sides of the tube, they would have coalesced to form droplets that would have entered the sputum since the tube was curved creating a gradient for the droplets to enter the sputum. Further, the treatment is uniform, and any error incurred would be uniform in all the treatments.

In my opinion, the dose of API introduced into the sputum with your kit may vary, especially due to the process of deposition of droplets on the walls. The error may not be critical and may be omitted from the calculations. Despite this, I still don't understand why it was decided to use nebulization in the endotracheal tube system instead of pipetting the drug into the sputum. Nebulization was not the focus of the research...

Response: 

The reason why we used an endotracheal tube to investigate the efficacy of aerosolised  BromAc is that in the first place, endotracheal tubes inserted in patients get clogged with thick sputa that is a result of excess sputum secreted in the lungs which settle in the tubes Further, the thick sputum settles in the tubes and is capable of clogging the tubes. In addition, the efficacy of BromAc in solubilising this sputum would indicate its efficacy in patients with respiratory diseases where a copious amount of thick purulent sputa are produced.

Delivery by nebulisation, in this case, is only a simulation of the real scenario where patients are treated using this method of delivery.

Reviewer 2 Report

To be scientifically sound, the paper should contain statistical analysis to confirm the diffeences.

Author Response

Dear reviewer,

We thank you for your revision of our manuscript and for your comment.

Statistical analysis of tables 2, 3 and 4 have been performed to compare control and test-drug samples. Table legends have been updated accordingly. Statistical analysis method has been added to the methods section.

Round 3

Reviewer 1 Report

The manuscript will be proper for publication after correction of the viscosity unit which are still wrong. I do not know how to explain this simple issue in the way that guarantee tahat authors will react. I repeat this again:  viscosity unit is Pa*s which means Pascal multiplied by second!!!!!!! NOT Pa alone which is unit for pressure or stress. In scientific publications it is important to use and understand the basics such as units of values used during investigations.  

Author Response

Dear reviewer,

We thank you for your revision of the manuscript.

The viscosity unit has been changed into [Pa * s ] on page 2 line 45, tables 2, 3 and 4.

Reviewer 2 Report

The authors found that the combination of the two drugs did not have a higher significancy as compared to single drug (see Tables 2, 3 and 4 based on statistical analysis). May you give an explanation?

Author Response

Dear reviewer,

Thank you for your comment.

In tales 2, 3, and 4 we compared test drugs versus control (vehicle).

We further analysed the significance of data by comparing single agents to the drug combination (Supplement table 2). The new analysis showed the significance of the drug combination over the single agents. Modifications are tracked change.